# Machine Learning Prediction of Comorbid Substance Use Disorders among People with Bipolar Disorder

**DOI:** 10.3390/jcm11143935

**Published:** 2022-07-06

**Authors:** Vincenzo Oliva, Michele De Prisco, Maria Teresa Pons-Cabrera, Pablo Guzmán, Gerard Anmella, Diego Hidalgo-Mazzei, Iria Grande, Giuseppe Fanelli, Chiara Fabbri, Alessandro Serretti, Michele Fornaro, Felice Iasevoli, Andrea de Bartolomeis, Andrea Murru, Eduard Vieta, Giovanna Fico

**Affiliations:** 1Bipolar and Depressive Disorders Unit, Institute of Neurosciences, Hospital Clinic, University of Barcelona, IDIBAPS, CIBERSAM, 170 Villarroel St., 12-0, 08036 Barcelona, Catalonia, Spain; voliva@clinic.cat (V.O.); mdeprisco@clinic.cat (M.D.P.); anmella@clinic.cat (G.A.); dahidalg@clinic.cat (D.H.-M.); igrande@clinic.cat (I.G.); amurru@clinic.cat (A.M.); gfico@clinic.cat (G.F.); 2Department of Biomedical and Neuromotor Sciences, University of Bologna, 40123 Bologna, Italy; giuseppe.fanelli5@unibo.it (G.F.); chiara.fabbri@yahoo.it (C.F.); alessandro.serretti@unibo.it (A.S.); 3Section of Psychiatry, Department of Neuroscience, Reproductive Science and Odontostomatology, Federico II University of Naples, 80131 Naples, Italy; dott.fornaro@gmail.com (M.F.); felice.iasevoli@unina.it (F.I.); adebarto@unina.it (A.d.B.); 4Addictions Unit, Department of Psychiatry and Psychology, Institute of Neuroscience, Hospital Clinic, University of Barcelona, IDIBAPS, CIBERSAM, 170 Villarroel St., 12-0, 08036 Barcelona, Catalonia, Spain; mtpons@clinic.cat (M.T.P.-C.); prguzman@clinic.cat (P.G.); 5Department of Human Genetics, Radboud University Medical Center, Donders Institute for Brain, Cognition and Behavior, 6525 GD Nijmegen, The Netherlands; 6Social, Genetic & Developmental Psychiatry Centre, Institute of Psychiatry, Psychology & Neuroscience, King’s College London, London SE5 9NU, UK

**Keywords:** bipolar disorder, substance use disorder, cannabis use disorder, alcohol use disorder, machine learning

## Abstract

Substance use disorder (SUD) is a common comorbidity in individuals with bipolar disorder (BD), and it is associated with a severe course of illness, making early identification of the risk factors for SUD in BD warranted. We aimed to identify, through machine-learning models, the factors associated with different types of SUD in BD. We recruited 508 individuals with BD from a specialized unit. Lifetime SUDs were defined according to the DSM criteria. Random forest (RF) models were trained to identify the presence of (i) any (SUD) in the total sample, (ii) alcohol use disorder (AUD) in the total sample, (iii) AUD co-occurrence with at least another SUD in the total sample (AUD+SUD), and (iv) any other SUD among BD patients with AUD. Relevant variables selected by the RFs were considered as independent variables in multiple logistic regressions to predict SUDs, adjusting for relevant covariates. AUD+SUD could be predicted in BD at an individual level with a sensitivity of 75% and a specificity of 75%. The presence of AUD+SUD was positively associated with having hypomania as the first affective episode (OR = 4.34 95% CI = 1.42–13.31), and the presence of hetero-aggressive behavior (OR = 3.15 95% CI = 1.48–6.74). Machine-learning models might be useful instruments to predict the risk of SUD in BD, but their efficacy is limited when considering socio-demographic or clinical factors alone.

## 1. Introduction

Substance use disorder (SUD) frequently occurs among people with bipolar disorder (BD), worsening their clinical trajectories [1,2]. A comorbid diagnosis of BD and SUD occurs in up to 30–60% of people with SUD, depending on the substance used, including alcohol [3], cannabis [4], tobacco [5], or others [3,6,7], with men having higher lifetime risks of SUD than women [8]. The presence of SUD accounts for a higher number of lifetime mood episodes and hospitalizations [9]; lifetime medical comorbidities [10]; reduced cognitive and psychosocial functioning [11]; and an increased risk for suicide [12], impulsive and aggressive behavior [13], or mortality [14]. Substance use may also attenuate the efficacy or compliance to psychopharmacological treatments, further worsening BD course [15,16].

The strongest comorbid associations of SUD among individuals with BD are found with alcohol use disorder (AUD), followed by cannabis and other illicit drugs [8]. Interestingly, the most current report by the National Epidemiological Survey on alcohol and related conditions [17] suggests that the presence of both alcohol use and having a psychiatric diagnosis, including BD, are associated with higher utilization rates of lifetime poly-substance abuse [18] compared with individuals without these clinical characteristics. Patients with BD with multiple SUDs have even more severe outcomes, including the risk of overdose, criminal conviction, low adherence to treatments, and reduced global functioning [10,19,20].

Despite its burden, the relationship between SUDs and BD has been minimally studied. Indeed, a few longitudinal studies have examined the predictors of SUD onset in BD, reporting that alcohol use disorder (AUD) might be predicted by psychotic symptoms [21], while cannabis use disorder might be predicted by younger age, lower education, and previous substance use [22]. In addition, the generalizability of much-published research on this issue is problematic, given that individuals who exclusively meet the criteria for a single SUD do not represent the naturalistic population in clinical settings [19]. This more significant symptomatic burden of comorbid SUDs in adults with BD points out the necessity of identifying the risk factors of co-occurrence in order to implement appropriate preventative strategies.

Evidence has suggested the feasibility of developing predictive models in psychiatry through machine-learning algorithms [23,24]. Several studies have used data mining and machine learning techniques to predict patient outcomes, including SUD [25]. However, to the best of our knowledge, no study has applied machine-learning techniques to date to predict the presence of comorbid SUD in individuals with BD. In addition, no studies on the topic have analyzed to what extent BD phenotypes differ according to the type of SUD.

The current study aims at identifying the most meaningful variables associated with SUD, AUD, and AUD in comorbidity with any other SUD in a large sample of patients with BD through the use of a random forest (RF) model. These variables will then be used in a regression model to provide further information on the associations between BD and specific types of SUD.

## 2. Materials and Methods

### 2.1. Participants

The present study included all the patients enrolled in the systematic prospective follow-up of the Bipolar Disorders Unit of the Hospital Clinic, University of Barcelona, Catalunia, Spain, from October 1998 to November 2021. Barcelona’s Bipolar and Depressive Disorders Unit provides both tertiary- and secondary-level care. The unit enrolls difficult-to-treat BD patients BD derived from all over Spain, and patients from a catchment area of approximately 170,000 inhabitants in Barcelona in particular [26]. Trained psychiatrists regularly treat more than 700 patients according to a local protocol based on international clinical guidelines [27,28]. The inclusion criteria were as follows: (i) older than 18 years of age, and (ii) diagnosis of BD type I (BDI) or type II (BDII) according to the Diagnostic and Statistical Manual of Mental Disorders (DSM) IV [29], DSM-IV-TR [30], or DSM-5 criteria [31]. In addition, the exclusion criteria were the presence of severe organic diseases requiring urgent treatment at baseline assessment or severe cognitive, motor, or visual impairment. All participants provided written informed consent for this ethical committee-approved study (approval code: HCB/2017/0432).

### 2.2. Clinical Variables Assessment

Patients were assessed using the Structured Clinical Interview for DSM Disorders [32]. The main sociodemographic and clinical characteristics were collected through an ad hoc schedule. If specific information was not collected during the baseline assessment, the electronic clinical records of each patient were inquired. Collected variables included age, education, working and living status, duration of illness, current pharmacological treatment (if maintained for at least six months), the number of hospitalizations and affective episodes, and lifetime aggressive behavior, among other variables of interest (see Table 1). Predominant polarity was defined according to the standard definition created in our unit and repeatedly validated [33]. A family history of psychiatric disorder was defined as having a first-degree relative diagnosed with and/or treated for any mood disorder, including major depression, cyclothymia, and dysthymia. The term “suicide attempt” refers to intentional self-inflicted poisoning, injury, or self-harm with a deadly intent without death. The presence of lifetime SUD was assessed according to DSM-IV [29], DSM-IV-TR [30], or DSM-5 criteria [31], including drug-specific diagnoses for ten substances: alcohol, cannabis, cocaine, heroin, hallucinogens, inhalants, prescription opioids, sedatives/tranquilizers, stimulants, and other drugs (e.g., ecstasy or ketamine). Each DSM-5 SUD diagnosis required positive responses to 2 or more of the 11 criteria for each drug-specific SUD.

### 2.3. Statistical Analyses

All of the analyses were conducted with RStudio, R version 4.1.2 [34]. The Kolmogorov–Smirnov test was used to assess whether continuous variables displayed a normal distribution. The parametric comparative analyses for the demographic and clinical characteristics of the groups (SUD vs. non-SUD) were done with unpaired t-test with Bonferroni post hoc correction for continuous variables; for non-parametric distributions, a Mann–Whitney *U* Test or Kruskal–Wallis Test was used, where appropriate. Categorical data were analyzed by Chi-square analysis.

#### 2.3.1. Missing Data

We inspected missing data using the R package “skimr” [35], and these were assumed to be missing at random. Therefore, we included only the variables presenting at least 75% of the available data in the model. For those contributing less than 25% of missing data, missing data were imputed using the R package “missRanger” [36], which is based on the algorithm of “missForest” (Stekhoven and Bühlmann 2012) and uses a random forest (RF) approach. The parameter “num.trees” was set at 5000, and the out-of-bag (OOB) errors were calculated for each variable in order to measure the accuracy according to the method outlined in previous evidence [37]. OOB errors ranged from 0 (better performance) to 1 (worse performance).

#### 2.3.2. Random Forest

We performed RFs using the R packages “RandomForest” [38] and “caret” [39] to tune the algorithm. A series of RFs were primarily used to select the most important features, which predicted different outcomes. To select the most critical variables, the “mean decrease of Gini coefficient” was adopted to observe which variables substantially contributed to the homogeneity of the nodes and leaves in the resulting RF.

**Table 1 jcm-11-03935-t001:** Socio-demographic and clinical characteristics of the sample classified according to the presence of substance use disorder(s).

	SUD (*n* = 262; 51.57%)	Non-SUD (*n* = 246; 48.4%)	t/Z/χ^2^	*p*
**Gender (*n*; %)**			35.32	<0.001
- Female	109; 41.6%	167; 67.9%		
**Living status (*n*; %)**			14.27	0.003
- Parents	45; 18.8%	83; 33.2%		
- Family	141; 59%	114; 45.6%		
- Alone	33; 18.8%	36; 14.4%		
- Other (community)	20; 8.4%	17; 6.8%		
**Relationship status (*n*; %)**			27.8	<0.001
- Not in a relationship	72; 29.8%	125; 48.4%		
- Married	117; 48.3%	84; 32.6%		
- Divorced	40; 16.5%	47; 18.2%		
- Widow	13; 5.4%	2; 0.8%		
**Working status (*n*; %)**			14.78	0.011
- Full-time or part-time job	127; 53.8%	131; 52.8%		
- Unemployed	27; 11.4%	23; 9.3%		
- Retired	33; 14%	45; 18.1%		
- Not able to work	34; 14.4%	17; 6.9%		
**Diagnosis (*n*; %)**			5.26	0.014
- BDI	190; 72.5%	155; 63%		
- BDII	72; 27.5%	91; 37%		
**Age and illness duration (mean ± SD)**				
Age at assessment	48.6 ± 15.08	43.77 ± 26.46	26.397	<0.001
Age at onset	29.3 ± 13.19	26.4 ± 10.63	28.307	<0.001
Duration of illness	19.27 ± 12.1	17.32 ± 12.06	29.959	0.048
**Number of affective episodes, lifetime (mean ± SD)**				
- Depressive	8.39 ± 11.87	7.13 ± 9.17	28.285	0.084
- Manic	2.25 ± 4.1	2.52 ± 3.69	35.186	0.066
- Hypomanic	5.51 ± 10.2	4.04 ± 7.76	28.288	<0.001
- Mixed	0.63 ± 1.89	0.65 ± 1.77	32.958	0.54
- Total	16.84 ± 22.61	14.34 ± 16.66	30.338	0.25
**Polarity of the first affective episode (*n*; %)**			7.94	0.09
- Depressive	168; 70.9%	66; 26.2%		
- Manic	52; 21.9%	22; 8.7%		
- Hypomanic	9; 3.8%	155; 61.5%		
- Mixed	6; 2.5%	5; 2%		
**Number of Psychiatric admissions, lifetime (mean ± SD)**	1.59 ± 2.14	1.61 ± 2.01	30.023	0.61
**Clinical course variables, lifetime (*n*; %)**				
- Suicide attempts	73; 51.4%	172; 47%	0.702	0.23
- Aggressive behaviours				
- Self-directed	52; 21%	52; 19.8%	0.130	0.4
- Hetero-directed	26; 10.6%	54; 20.6%	9.64	0.001
- Psychotic symptoms	119; 49.2%	132; 51.2%	0.198	0.36
- Rapid cycling	60; 22.9%	68; 27.6%	1.51	0.130
- Seasonality	63; 25.6%	53; 20.2%	2.08	0.09
- Family history of Mood Disorder	143; 62.2%	153; 61.4%	0.027	0.47
- Comorbidity with Personality Disorder			7.31	0.063
- Cluster A	4; 1.7%	7; 2.9%		
- Cluster B	20; 8.7%	34; 13.9%		
- Cluster C	11; 4.8%	4; 1.6%		

BD = bipolar disorder; SUD = substance use disorder; *n* = number of cases; *p* = statistical significance; SD = standard deviation; χ^2^ = Chi-square test; t = Independent Samples *t*-test; Z = Mann–Whitney U test.

We conducted a first RF considering the presence of lifetime SUD as the predicted condition in the entire sample. Next, among people with SUD, as the first step, we selected only people with a lifetime diagnosis of AUD who consumed alcohol alone or combined with other substances (i.e., cocaine, cannabis, hallucinogens, or MDMA). Then, we conducted a second and third RF considering the presence of AUD mono-use or comorbid with another SUD as the predicted conditions, compared with non-abusing controls; finally, we performed a fourth RF to differentiate people with AUD mono-use from people with AUD+SUD. RF is a classification algorithm that combines multiple decision trees made by randomly selected bootstrap samples, mainly affected by unbalanced data. We used the R package “ROSE” [40] to under sample the predicted class that presented the most observations in order to obtain a balanced dataset. As a secondary step, to exploratory assess the accuracy of the prediction model, a “train” and a “test” dataset were prepared, containing 80% and 20% of the original observations, respectively. We initially used a repeated cross-validated RF (10-folds, ten repeats) on the “train” dataset and then tuned some hyperparameters (i.e., number of trees, number of features randomly selected at each node, and size of the node) during the learning phase, to obtain the best accuracy value. Then, we applied the trained model to the “test” dataset, and calculated measures such as accuracy, sensitivity, specificity, the OOB estimate, and the F-score (F1), which alternatively are tests of the model’s accuracy. To graphically present our results, we produced a confusion matrix.

#### 2.3.3. Multiple Logistic Regression

The variables selected by the RFs that mostly decreased the homogeneity of the nodes and leaves of the model were considered as independent variables in multiple logistic regressions, adjusting for sex, duration of illness, and age at BD onset, as these are known factors associated with SUD in previous studies [6,20,41]. The dependent variables were the same ones considered in the corresponding RFs. Odds ratios (ORs) and their 95% confidence intervals (CIs) were estimated to assess the significance of each result. The variance explained was calculated as Nagelkerke’s pseudo R^2^ with the R package “fmsb” [42]. These regression models were useful to provide further information on the association between the relevant variables identified by the RFs and the various SUD comorbidities of interest.

## 3. Results

### 3.1. Characteristics of the Sample

A total of 508 patients were included, of which 276 were female (54.3%). The mean age of the total sample was 46.11 (standard deviation—SD = 14.47) years old. Our sample consisted of 345 (67.9%) patients with BD-I and 163 (32.1%) patients with BD-II. Of all of the patients, 262 (51.57%) fulfilled the DSM criteria for lifetime SUD of any type. The most used substance was alcohol (42.1%), followed by cannabis (22.6%), cocaine (12%), amphetamine (4.7%), MDMA (4.7%), and hallucinogens (2.1%). A total of 106 patients (20.8%) had AUD with at least another SUD (AUD+SUD). The sample characteristics are presented in Table 1.

### 3.2. Missing Data

Among the variables presenting missing data, 22 of them presented less than 25% of missing values. Of these, 17 presented fewer than 10% of missing values. Errors estimated during the imputation of missing data were less than 20%, except for the number of lifetime hospitalization (OOB = 0.29).

### 3.3. Patients with SUD vs. without SUD

The RF model performance outputs are reported in Table 2. The variables presenting higher values of mean decrease of gini were “number of total affective episodes”, “number of total depressive episodes”, “number of total hypomanic episodes”, “number of total manic episodes”, “number of lifetime hospitalization”, “being in a relationship”, “diagnosis of cluster B personality disorder”, “number of attempted suicides”, “number of mixed episodes”, and “treatment with benzodiazepines”.

These variables were tested in a multiple logistic regression adjusted for relevant covariates (see Methods). The presence of SUD was positively associated with a diagnosis of cluster B personality disorder (OR = 2.31 [95% CI = 1.26–4.23]; *p* = 0.006), and negatively associated with being in a relationship (OR = 0.6 [95% CI = 0.39–0.91]; *p* = 0.015) (Figure 1). The model explained 16.9% of the total variance in the sample of BD with SUD vs. non-SUD.

### 3.4. Patients with AUD vs. without SUD

RF model performance outputs are reported in Table 2. The variables presenting higher values of mean decrease of gini were “number of total affective episodes”, “number of total depressive episodes”, “number of total hypomanic episodes”, “number of total manic episodes”, “number of mixed episodes”, “number of lifetime hospitalization”, “number of attempted suicides”, “being in a relationship”, “diagnosis of cluster B personality disorder, any”, “treatment with mood stabilizers other than lithium”.

In a multiple logistic regression model adjusted for relevant covariates (see Methods), none of these variables was significantly associated with the presence of AUD (Figure 1). The model explained 14.2% of the total variance.

**Table 2 jcm-11-03935-t002:** Random forest model performance outputs.

Data Set	Number of Trees	Number of Features	Node Size	Accuracy%	95% CI	*p*	Sensitivity	Specificity	F1-Score
SUD VS. NO-SUD	800	9	6	65.3%	54.8–74.7	0.004	69.6%	61.2%	0.66
AUD VS. NO-SUD	350	3	43	53.8%	39.5–67.8	0.44	44%	63%	0.48
AUD+SUD VS. NO-SUD	500	26	47	75%	56.6–88.5	0.003	75%	75%	0.75
AUD+SUD VS. AUD	800	2	14	62.5%	43.6–78.9	0.107	43.8%	81.2%	0.54

Substance use disorder: SUD; alcohol use disorder (AUD); confidence interval (CI).

### 3.5. Patients with AUD+SUD vs. without SUD or AUD

RF model performance outputs are reported in Table 2. The variables presenting higher values of mean decrease of gini were “number of total hypomanic episodes”, “presence of hetero-directed aggressivity”, “number of total affective episodes”, “number of total depressive episodes”, “violent suicide attempt”, “number of total manic episodes”, “being in a relationship”, “number of lifetime hospitalization”, “first episode as hypomanic”, “presence of melancholia”.

In a multiple logistic regression adjusted for relevant covariates, hypomania as the first affective episode (OR = 4.34 [95% CI = 1.42–13.31]; *p* = 0.01) and hetero-directed aggressivity (OR = 3.15 [95% CI = 1.48–6.74]; *p* = 0.003) were associated with AUD+SUD (Figure 1). The model explained 31.5% of the total variance in AUD+SUD.

### 3.6. Patients with AUD+SUD vs. with AUD

RF model performance outputs are reported in Table 2. The variables presenting higher values of mean decrease of gini were “number of total affective episodes”, “number of total manic episodes”, “number of total depressive episodes”, “number of total hypomanic episodes”, “mood disorders familiarity”, “number of lifetime hospitalization”, “number of total mixed episodes”, “first episode as depressive”, “presence of rapid-cycling”, and “atypical depression”.

These variables were considered in an adjusted multiple logistic regression. The presence of another SUD in the context of AUD was negatively associated with having depression as the first affective episode (OR = 0.41 [95% CI = 0.21–0.81]; *p* = 0.011) (Figure 1). The model explained 30.5% of the total variance.

**Figure 1 jcm-11-03935-f001:**
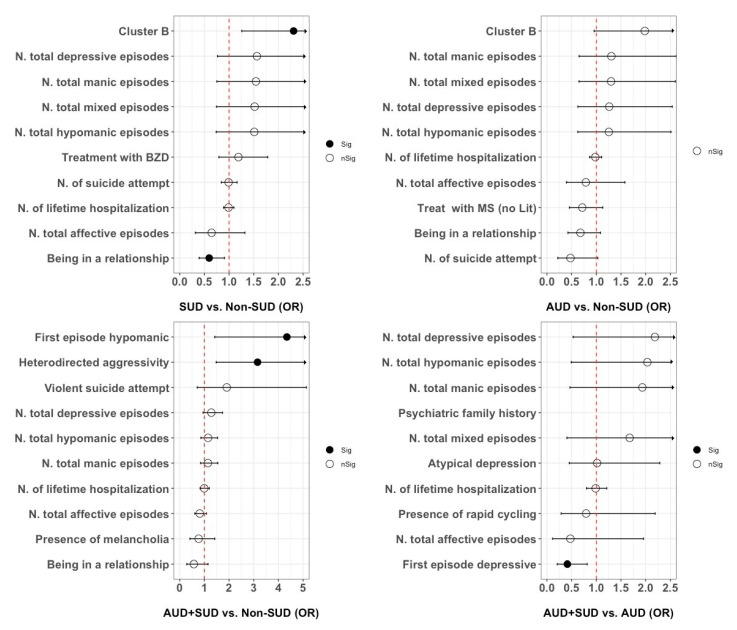
Logistic regression plots of odds ratio (OR) and 95% confidence intervals (CI). Independent variables were selected among the top ten features derived from the random forest (RF) models. The four models predicted (from left to right): any substance use disorder (SUD) in the total sample, alcohol use disorder (AUD) in the total sample, AUD co-occurrence with at least another SUD in the total sample, and AUD co-occurrence with at least another SUD among BD patients with AUD.

## 4. Discussion

To the best of our knowledge, this is the first study attempting to examine the value of socio-demographic and clinical factors for the prediction of SUD in a large, naturalistic sample of adults with BD using a machine-learning approach.

Using a random forest classifier, we developed models to predict the presence of SUD, AUD, or the co-occurrence of AUD and other SUDs in BD. Although the specificities of the models were acceptable, their accuracies were low to moderate. The comparison of the performance of our models with previously developed models is limited by the scarce evidence on the topic [20].

The model with the highest accuracy was the one predicting the co-occurrence of another SUD among individuals with AUD and compared with those without SUD, correctly classifying up to 75% of the sample. The top features were similar among the four random forest models and included clinical factors associated with a severe course of illness in BD, such as the lifetime number of affective episodes and the respective episode polarity [43], type of index episode [44], the presence of comorbid cluster B personality disorder [45], and suicide or aggressive behavior [46]. It should be remarked that the top features extracted might be highly correlated with other relevant variables associated with poor BD outcomes (e.g., presence of psychotic symptoms, and low socioeconomic status) [47,48], thus hindering their effect on SUD prediction. However, the effect of other variables could be considered minimal compared with that of the selected top features [49]. While we found similarities among the top features identified by the RF models, their association with SUD comorbidities varied among the logistic regression models.

A lifetime comorbid diagnosis of cluster B personality disorder and not being in a relationship predicted the presence of SUD vs. no-SUD. This result is not surprising as SUD, cluster B personality disorders, and BD are characterized by impulsivity and poor behavioral control [50,51,52]. The complex phenotypic overlap between BD and cluster B personality disorders is a clinical challenge [53], with problematic clinical and genetic boundaries [54], frequently leading to a misdiagnosis of BD in people with personality disorders, such as borderline personality disorder (BPD) [55,56]. The risk of substance use and abuse in individuals with BD and comorbid BPD is two to three times higher than in individuals with BD alone [57]. This could be possibly justified by an even higher tendency toward risky behaviors, mood instability, impulsivity, affective reactivity, and context-specific increased sensitivity to rewards in patients with comorbid BD and BPD, ultimately leading to substance misuse [58]. Another variable associated with SUD risk is the lack of a stable relationship, which is in line with previous evidence [59]. Similarly, socio-economic functioning is substantially decreased in patients with BD, with lower odds of being in a stable relationship compared with the general population (Sletved et al., 2021 [60]), while social or family support improves patients’ global functioning [61].

Given the extremely high prevalence of AUD in BD and their strong interplay, we analyzed predictors of AUD alone or comorbid with another SUD. The relationship between AUD and BD is complex and comprises shared biological pathways [62], as well as clinical and psychological characteristics [63]. However, previous observational studies on AUD in BD were mainly conducted on individuals with a co-occurrence of other SUDs [3,21,64], without clinical phenotyping, based on a distinct pattern of use. After controlling for sex, age of onset, and duration of illness, no other factors were associated with AUD without other SUDs in our sample. However, AUD comorbid with another SUD was positively associated with a history of a hypomanic episode at BD onset and hetero-aggressive behavior compared with non-use, and negatively associated with a history of a depressive episode at BD onset when compared with non-use. The polarity of the first episode has a relevant influence on the course of BD, with the depressive one being the most common and being related to suicide attempts [65], with (hypo)manic being related to alcohol or other substance misuses [44]. Given that polarity at onset might predict subsequent predominant polarity in BD [44,66], its evaluation may guide long-term therapeutic planning [67]. The link between first-episode polarity or predominant polarity, SUDs, and BD requires further analysis in prospective longitudinal studies, as affective episodes may be triggered by substance use, thus influencing lifetime affective episodes of a specific polarity [68]. Aggressive behavior is considered a trait and a state factor associated with BD, often driven/worsened by substance use [52]. Proneness to impulsivity may lead to greater involvement in substance use and an increased risk for criminal, violent, or aggressive acts. However, these premises and the existence of putative common biological underpinnings of aggressive behavior and BD suggest that this undesirable outcome might result from environmental–gene interactions [69].

Individuals that reported substance misuse before the onset of BD are sometimes considered to have a “milder” BD phenotype [70]. In addition, sub-threshold mood symptoms or mood instability might be the result of substance use and might lead to BD misdiagnosis [71]. Therefore, the direction of the association between SUD and BD is relevant, as it might depict two different subpopulations of individuals according to the onset of SUD (i.e., before or after BD onset) with distinct clinical needs. However, our study lacked information about the differences between these subpopulations and the direction of the association. Several other limitations in this study should be considered. The cross-sectional design of the study, as well as the use of clinical variables, collected retrospectively from electronic clinical records, may have affected the accuracy and reliability of our data, particularly regarding previous affective episodes, hypomanic onsets—for which retrospective diagnosis is a clinical challenge—or mixed episodes—for which the DSM definition varied across the years. Obviously, a longitudinal study would be a better design to test our models [72]. In addition, we only included data on current psychopharmacological treatment, but not on psychosocial, psychoeducational, or other psychological interventions that are highly recommended for comorbid SUD management in major guidelines [73] because they improve adherence to pharmacological treatment, leading to a more stable BD course [74]. Secondly, SUD might have been underdiagnosed because of internalized stigma [75]. Given that patients were recruited from a specialized unit, a potential selection bias should also be taken into account, as we could assessed the most severe cases that were ultimately forwarded to a tertiary clinic or, conversely, the less severe ones. Furthermore, when considering lifetime SUD, we might have excluded people with current SUD, thus inflating the risk for Berkson’s bias, and ultimately reducing the overall generalizability of the results. Finally, RFs are a “black box”, making any local interpretation of a specific prediction quite impractical.

Despite its possible limitations, the present study is the first one to develop algorithms to identify SUD in patients with BD and to describe potential sociodemographic and clinical predictors of comorbidity. Furthermore, our data come from a highly specialized unit, in which patients are regularly followed-up by trained psychiatrists.

## 5. Conclusions

Bipolar disorder that occurs in comorbidity with substance use disorder represents a severe clinical phenotype of bipolar illness. Alcohol use disorder is the most frequent comorbid substance use disorder in individuals with bipolar disorder, and it frequently presents in co-occurrence with other substance use. Machine-learning models might be used to predict the risk of having a comorbid substance use disorder in bipolar disorder, but their accuracy is limited when they include socio-demographical or clinical factors alone, as done in this study. Complex models integrating biological and clinical predictors represent a promising alternative to improve the performance of prediction.

## Data Availability

Data are available upon request.

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
