# Peer review of "Machine Learning Prediction of Comorbid Substance Use Disorders among People with Bipolar Disorder"

_jcm, 2022, doi:10.3390/jcm11143935_

Round 1
Reviewer 1 Report
This is a very interesting and innovative study, the machine learning method is very sophisticated, conclusions are clinically and scientifically important. However, one point needs explanation - the possible influence of the treatment methods, it should be analysed or described in more details.
Author Response
We agree with Reviewer 1 that for intermediate and long-term treatment of comorbid SUD in severe mental disorders, the question persisted whether SUD needs to be treated first and sufficiently before attention should be paid to the mental health disorder. Today, strategies that promote concomitant therapy of dual disorders are the established treatment of choice and recommended in major guidelines. In this perspective, psychotherapy, psychoeducation, or other psychological interventions integrated with psychopharmacological treatments may assure better outcomes, since psychological intervention may also contribute to improving adherence to pharmacological treatments.
Unfortunately, we did not recollect data on psychological interventions for some key reasons: these interventions are sometimes heterogeneous in time and type lacking standardization; only some of these interventions are currently available in the public healthcare system with a high number of patients using different psychological interventions in private settings.
We are aware that an integrated approach is the first option in comorbid SUD in BD, so we added a specific paragraph in the limitation section to address this important issue.

Reviewer 2 Report
The current study aims at identify clinical and sociodemographic factors associated with substance use disorder (SUD) and alcohol use disorder (AUD), in individuals with bipolar disorder (BD), using machine learning models. An interesting finding is the association of hypomania as first affective episode and hetero-aggressive behavior with AUD and SUD. The major strengths of the study are the large sample size (N=508) and the use of the RandomForrest algorithm. This approach is increasingly used in psychiatry and addictology. The obtained performances are average but, as indicated by the authors, they will probably be improved with the incorporation of biological data. The manuscript is well written and easy to read.
Minor concerns
Please include the Clinical Trial number of the clinical study in the material and methods section
Author Response
We are grateful to Reviewer 2 for his/her commentaries. We revised the manuscript in track change mode according to this suggestion.
